# Strong light-matter coupling for reduced photon energy losses in organic photovoltaics

Vasileios C. Nikolis [1,5], Andreas Mischok [2], Bernhard Siegmund [1,5], Jonas Kublitski [1], Xiangkun Jia[1], Johannes Benduhn [1], Ulrich Hörmann[3], Dieter Neher [3], Malte C. Gather [2], Donato Spoltore [1] & Koen Vandewal [1,4]

Strong light-matter coupling can re-arrange the exciton energies in organic semiconductors. Here, we exploit strong coupling by embedding a fullerene-free organic solar cell (OSC) photo-active layer into an optical microcavity, leading to the formation of polariton peaks and a red-shift of the optical gap. At the same time, the open-circuit voltage of the device remains unaffected. This leads to reduced photon energy losses for the low-energy polaritons and a steepening of the absorption edge. While strong coupling reduces the optical gap, the energy of the charge-transfer state is not affected for large driving force donor-acceptor systems. Interestingly, this implies that strong coupling can be exploited in OSCs to reduce the driving force for electron transfer, without chemical or microstructural modifications of the photo-active layer. Our work demonstrates that the processes determining voltage losses in OSCs can now be tuned, and reduced to unprecedented values, simply by manipulating the device architecture.

---

[1] Dresden Integrated Center for Applied Physics and Photonic Materials (IAPP) and Institute for Applied Physics, Technische Universität Dresden, Nöthnitzer Str. 61, 01187 Dresden, Germany. [2] Organic Semiconductor Centre, SUPA, School of Physics and Astronomy, University of St. Andrews, North Haugh, St, Andrews KY16 9SS, UK. [3] Institute of Physics and Astronomy, University of Potsdam, Karl-Liebknecht-Straße 24-25, 14476 Potsdam, Germany. [4] Institute for Materials Research (IMO-IMOMEC), Hasselt University, Wetenschapspark 1, 3590 Diepenbeek, Belgium. [5] Present address: Heliatek GmbH, Treidlerstraße 3, 01139 Dresden, Germany. Correspondence and requests for materials should be addressed to V.C.N. (email: vasileios_christos.nikolis@iapp.de) or to A.M. (email: am470@st-andrews.ac.uk) or to K.V. (email: koen.vandewal@uhasselt.be)

Organic solar cells (OSCs) based on electron donating (D) and electron accepting (A) materials nowadays exceed power conversion efficiencies (PCE) of 16% in single cells, and 17% in tandem configuration[1–3]. While recent progress has been impressive, OSCs continue to suffer from rather large voltage losses. For any photovoltaic technology, it is important that the energy loss related to the photon-to-electron conversion process is minimized, providing the highest possible photovoltage. A lower limit for the energy lost during photon-conversion is given by the difference between the optical gap of the solar cell ($E_{opt}$) and $qV_{OC}$. Here, $V_{OC}$ is the open-circuit voltage of the device under 1 sun illumination and $q$ the elementary charge. This difference is often more than 0.6 eV for OSCs, which is 0.2 to 0.3 eV higher than for silicon, gallium arsenide, or perovskite-based solar cells[4]. Recent work has shown that one important reason for this large loss is their relatively large non-radiative decay rate as compared to the radiative one[5–7]. Moreover, organic photovoltaic materials typically exhibit shallower absorption band tails than their inorganic counterparts, due to energetic disorder, molecular vibrations and the presence of low-energy intermolecular charge-transfer (CT) states[8,9]. The shallow absorption edge in the energy range around $E_{opt}$ can induce rather large losses, by shifting the strong absorption to photon energies significantly higher than $E_{opt}$. Generally, the absorption edge of a solar cell should be as steep as possible, mimicking the ideal step-wise absorption spectrum required for reaching the efficiency upper limit as predicted by Shockley and Queisser[10,11]. As OSC efficiencies are currently limited by the large voltage losses, there is an urgent need for strategies to reduce the $E_{opt}-qV_{OC}$ losses and steepen the absorption edge to improve the PCE of organic photovoltaic devices.

Due to the use of (at least) one highly transmissive contact and the high absorbance in their active layers, OSCs typically form weak optical cavities[12]. However, when embedding the organic materials into a planar Fabry–Pérot optical cavity which is on resonance with their electronic transitions, strong or ultra-strong exciton–photon coupling and polariton formation can occur at room temperature as a result of their large oscillator strength[13–19]. In the strong coupling regime, cavity photons and excitons hybridize into light–matter excited states (polaritons), which have different energies than the initial uncoupled states[20]. Among others, strong coupling in organic semiconductors has been reported to lead to longer exciton diffusion lengths[21–23], higher charge carrier mobilities[24], control of photoisomerization[25], extended responsivity[26], and enhanced intersystem crossing between singlet and triplet states[27].

In this work, we explore a strategy to reduce photon energy losses in OSCs, through the use of strong light–matter coupling (SC). Hereby, we induce new states which exhibit a red-shifted $E_{opt}$ as compared to the pristine absorbers. The crucial point is that we benefit from an unchanged $V_{OC}$, by bringing the device's $E_{opt}$ closer to $V_{OC}$. As showcase systems, we use chloroboron subnaphthalocyanine (SubNc) as donor and hexachloro phenoxy subphthalocyanine (Cl$_6$-PhOSubPc) or Buckminster fullerene (C$_{60}$) as acceptors. We substitute the highly transmissive indium tin oxide (ITO) bottom electrode with a more reflective silver (Ag) mirror-electrode forming a high-quality optical cavity (Fig. 1a). In both SubNc/Cl$_6$-PhOSubPc and SubNc/C$_{60}$ cells we observe the formation of polaritons, accompanied by the splitting of singlet absorption peaks, resulting in a redshifted absorption, which lowers the device's $E_{opt}$. For the higher voltage SubNc/Cl$_6$-PhOSubPc device, we achieve a $V_{OC}$ of 1.151 V. By tuning the device cavity thickness, the polariton absorption redshifts, reducing $E_{opt}$ by 56 meV, from 1.727 to 1.671 eV. Accordingly, the energy losses $E_{opt}-qV_{OC}$ are effectively lowered from 0.576 eV down to 0.525 eV. To probe the effect of SC on the energetics of CT-states, we fabricated SubNc/C$_{60}$ devices, which exhibit more pronounced CT-features in absorption and emission. We find that the SubNc/C$_{60}$ CT-states do not undergo strong coupling, and $E_{CT}$ remains unchanged. This finding implies that in this device, the red-shifted $E_{opt}$ results in a reduction of the driving force for electron transfer as we increase the device thickness. The peak external quantum efficiency (EQE) values however, remain largely unaffected. Finally, the absorption edge of the photovoltaic device at energies close to $E_{opt}$ is steepened by SC. Starting from an Urbach energy ($E_U$) of 22.4 meV for the reference (non-strongly coupled) SubNc/Cl$_6$-PhOSubPc solar cell, $E_U$ is reduced to 15.6 meV in the SC-devices. This value is comparable to $E_U$ values observed for lead halide perovskites, and unprecedented for organic absorbers.

## Results

**Strong coupling in SubNc/Cl$_6$-PhOSubPc-based solar cells**. To support strong exciton–photon coupling, organic materials need to exhibit strong absorption and a significant overlap between absorption and emission, which is most easily achieved in materials with a sharp absorption edge and small Stokes shift. SubNc (donor) and Cl$_6$-PhOSubPc (acceptor) are ideal candidates to demonstrate SC in OSCs, since they both exhibit strong absorption at photon energies close to their $E_{opt}$, and Stokes shifts of only 16 nm (40 meV) and 25 nm (86 meV), respectively (Supplementary Fig. 1). The peak absorption ($E_P$) of SubNc is at 1.769 eV (701 nm, Fig. 1) and its $E_{opt}$ equals to 1.727 eV (717 nm). The combination of SubNc and Cl$_6$-PhOSubPc in an organic photovoltaic device leads to a PCE = 4.7% with a $V_{OC}$ = 1.151 V (Fig. 1 and Table 1). This leads to $E_{opt}-qV_{OC}$ losses of 0.576 eV and $E_P-qV_{OC}$ of only 0.618 eV, which are among the lowest for OSCs, implying already minimized voltage losses for this device.

For the SC-devices, we use a 25-nm-thick Ag bottom contact instead of ITO, and a 100-nm-thick Ag top contact. The cavity resonance wavelength $\lambda_{res}$ is mainly determined by the refractive index $n$ of the organic layers and the optical cavity length $t_{cav}$, i.e. the distance between the two Ag electrodes[28]. We tune $\lambda_{res}$ by varying the thickness of the electron transport layer (ETL) and hole transport layer (HTL), and we keep the active layer positioned in the field maximum in the center of the cavity by maintaining an equal thickness ($d$) for both HTL and ETL (Fig. 1a).

In Fig. 1b, the normalized EQE spectra of the reference and an exemplary SC-device (with $d = 55$ nm) reveal how SC affects the absorption peaks of SubNc and Cl$_6$-PhOSubPc. Compared to the EQE spectrum of the reference device, which shows two distinct peaks related to the absorption of SubNc and Cl$_6$-PhOSubPc (700 and 595 nm, respectively), the EQE spectra of the SC-devices are severely distorted. The absorption edge is steepened, while a redistribution of exciton energies occurs resulting in splitting of the material's absorption peaks. Among the new peaks, the low-energy one (743 nm, 1.669 eV) is significantly lower than the first singlet exciton peak of SubNc (717 nm, 1.729 eV), red-shifting the $E_{opt}$ of the device (Fig. 1d).

In order to extract the origin of the absorption redshift and new peaks in the SC-device, as well as to extract the coupling strengths, we perform a variation of the transport layers' thickness $d$. Figure 2a shows the EQE spectra of five SC-devices, selected out of a set of 10 devices used in the analysis described below (Supplementary Fig. 2), spanning the whole $d$ variation range from 31 to 55 nm. The resulting EQE spectra were compared to transfer-matrix (TM) simulations of the active layer

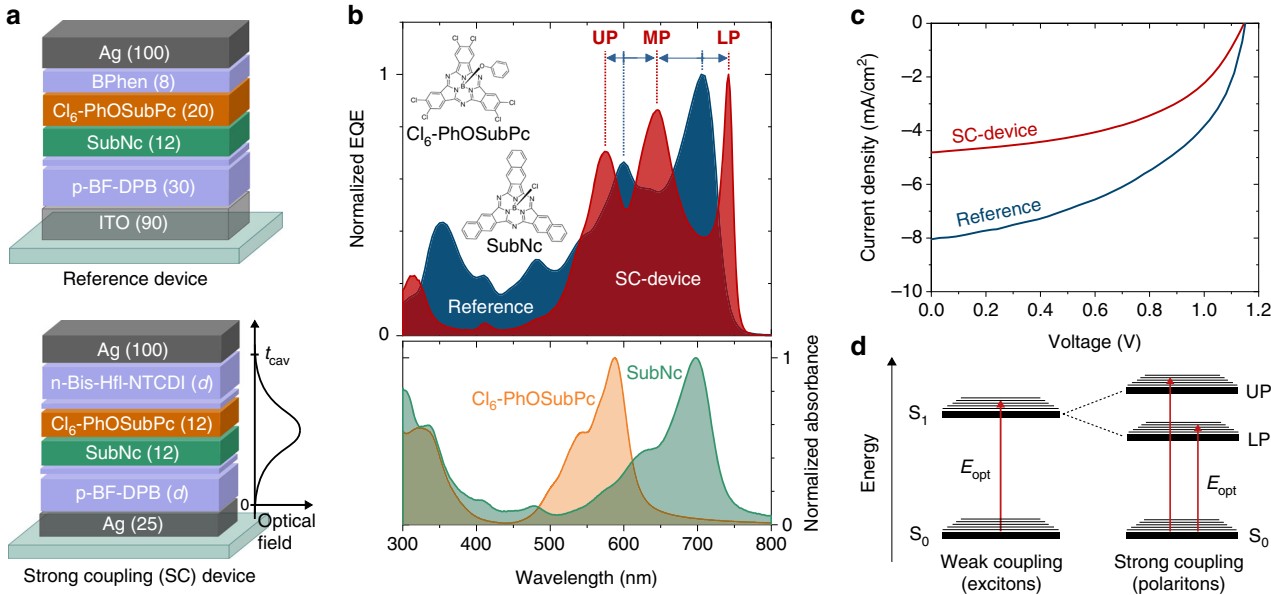

**Fig. 1** Effects of strong coupling (SC) on the performance of SubNc/Cl₆-PhOSubPc organic solar cells. **a** Device structure of a normal SubNc/Cl₆-PhOSubPc solar cell employing ITO as bottom contact used as reference, and general device structure of solar cells exhibiting SC effects. For the SC-devices, the thickness $d$ of the n-doped electron transport layer and p-doped hole transport layer are kept the same. **b** Normalized EQE spectra (upper panel) of the reference and an exemplary SC-device (with $d = 55$ nm) demonstrating the splitting of the absorption peaks of SubNc and Cl₆-PhOSubPc (lower panel). The blue solid arrows indicate splitting of the absorption peaks of both materials into upper-polariton, middle-polariton, and lower-polariton (UP, MP, and LP, respectively). As a result, the EQE spectrum of the SC-device is redshifted and its low-energy edge steepened. Inset pictures show the molecular structures of SubNc and Cl₆-PhOSubPc. **c** Current–voltage characteristic curves of the reference and SC-device show that the $V_{OC}$ remains rather constant under SC. **d** Excited-state diagram illustrating the splitting of the first singlet excited state ($S_1$) into two polariton states. Electronic transitions (red arrows) can occur directly from the ground state ($S_0$) to the high-energy upper polariton (UP) and the low-energy lower polariton (LP). The energetically lower LP defines the optical gap ($E_{opt}$) of the polariton based solar cell

**Table 1 Effect of strong coupling (SC) on the photovoltaic parameters of SubNc/Cl₆-PhOSubPc-based cells**

| Device | $V_{OC}$ (V) | $J_{SC}$ (mA cm⁻²) | FF (%) | PCE (%) | $E_{opt}$ (eV) | $E_{opt}$-$qV_{OC}$ (eV) |
|---|---|---|---|---|---|---|
| Reference | 1.151 | 7.7 | 53.5 | 4.7 | 1.727[a] | 0.576 |
| SC-device ($d = 55$ nm) | 1.146 | 4.8 | 50.0 | 2.8 | 1.671[b] | 0.525 |

Comparison between a reference device under no strong coupling, and an exemplary SC-device with $d = 55$ nm. Strong coupling does not affect the $V_{OC}$ of the device, while the red-shifting of the $E_{opt}$ leads to a reduction of the $E_{opt}$-$qV_{OC}$ losses
[a]Obtained as the crossing point of the device's EQE and EL spectra
[b]Obtained as the peak of the lower polariton branch

absorption, and modeled with the following coupled oscillator (CO) Hamiltonian:

$$\begin{pmatrix} E_P(\theta) & G_{PX_{SubNc}}/2 & G_{PX_{SubPc}}/2 \\ G_{PX_{SubNc}}/2 & E_{X_{SubNc}} & 0 \\ G_{PX_{SubPc}}/2 & 0 & E_{X_{SubPc}} \end{pmatrix} \begin{pmatrix} F_P \\ F_{X_{SubNc}} \\ F_{X_{SubPc}} \end{pmatrix} \\ = E_{Pol}(\theta) \begin{pmatrix} F_P \\ F_{X_{SubNc}} \\ F_{X_{SubPc}} \end{pmatrix} \quad (1)$$

where $E_i$ are the energies of the cavity photon (P) and the main SubNc ($X_{SubNc}$) and Cl₆-PhOSubPc ($X_{SubPc}$) excitons, $\theta$ is the angle of incidence, $G_{PX_{SubNc}}$ and $G_{PX_{SubPc}}$ are the photon–exciton coupling strengths, and $F_i$ are the photon and exciton fractions of the polariton. By simulating an empty cavity (replacing the absorber layers with non-absorbing spacers), we can extract the photon dispersion for independently varying both angles of incidence, and transport layer thicknesses. Figure 2b shows (i) the TM of the cavity, (ii) the solutions of the coupled oscillator model

and (iii) the extracted experimental EQE peak wavelengths from a series of devices (Supplementary Fig. 2). We observe excellent agreement between the optical TM simulation, the CO model and our experimental results, showing that excitons in both Cl₆-PhOSubPc and SubNc strongly couple to the cavity photons. One can clearly distinguish three polariton branches, the upper polariton (UP) below 595 nm, the middle polariton (MP) between 595 and 700 nm and the lower polariton (LP) above 700 nm. These polaritons are a mixture between $X_{SubNc}$, $X_{SubPc}$, and P, however only the MP has a significant percentage of both $X_{SubNc}$ and $X_{SubPc}$, while the LP mainly consists of $X_{SubNc}$ and P and the UP of $X_{SubPc}$ and P, respectively (Supplementary Fig. 3). The characteristic angular dispersion of the devices provides additional confirmation of the presence of SC, while the peak wavelengths in the reference device are angle independent. The coupling strengths can be extracted from the CO model, giving $G_{PX_{SubNc}} = 278$ meV and $G_{PX_{SubPc}} = 254$ meV. Comparing these results to the reference device with a transparent oxide electrode, we observe no dependence of the peak absorption wavelengths on the transport layer thicknesses (Supplementary Fig. 4).

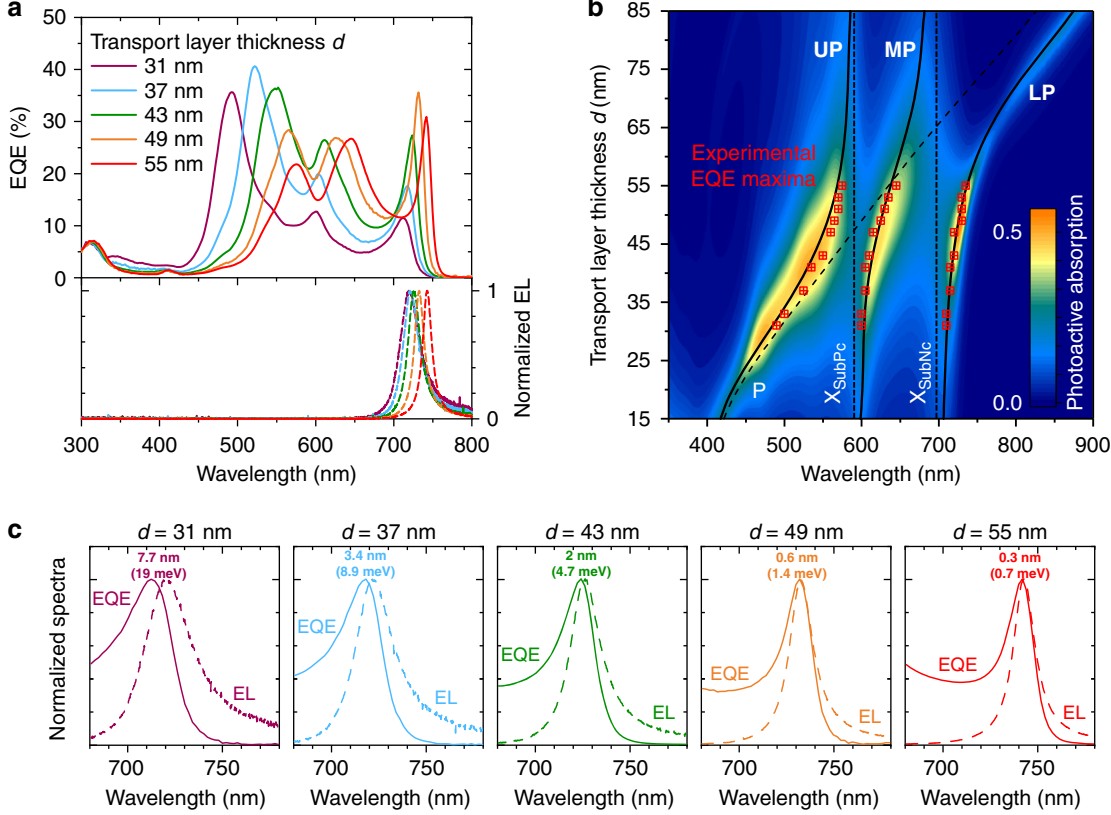

**Fig. 2** Thickness-dependent EQE and EL of SubNc/Cl$_6$-PhOSubPc-based devices under strong coupling. **a** EQE (upper panel) and normalized electroluminescence (EL) spectra (lower panel) of five SC-devices, for different transport layers' thickness $d$, showing the redshift of the device's absorption and emission for increasing $d$. Different values of $d$ lead to different cavity lengths and resonance wavelengths for the cavity photons. The five devices were exemplarily selected for the investigated range $d$, and the full series of devices is shown in Supplementary Fig. 2. **b** Simulated photoactive absorption (false color) in cells with varying $d$ showing the formation of polariton branches with pronounced anti-crossing at overlapping points of the cavity and exciton resonances (dashed lines). The simulation results agree well with the calculated polariton branches (solid lines) of a coupled oscillator model and the experimental EQE data (red squares). **c** Comparison between EQE and EL peaks of the LP peak for various $d$. The colored numbers (in nm and meV) denote the Stokes shift in each case. The minimal Stokes shifts confirm that the investigated devices operate in the strong coupling regime

In the reference device with ITO, $E_{opt} = 1.727$ eV corresponds to the energy of the first singlet excited state of SubNc, determined by taking the crossing point of the normalized reduced absorption and emission spectra of a solar cell employing only SubNc as photoactive material (Supplementary Fig. 5). Therein, absorption and emission peak wavelengths do not coincide as a consequence of the inhomogeneous broadening due to electron–phonon coupling and molecular vibrations. In contrast, a perfect polariton state would have no Stokes shift and be only a homogeneous linewidth, since the individual vibrational levels collectively couple to the optical transition[29]. Thus, the optical gap of SC-devices can be considered as the energy of the absorption or emission peak of the LP branch ($E_{peak,LP}$). Hence, the $E_{opt}$ of our exemplary SC-device ($d = 55$ nm) equals to $E_{opt} = E_{peak,LP} = 1.671$ eV. Minimal Stokes shifts of less than 1 nm are observed for our SC-devices when comparing the EQE and electroluminescence (EL) peaks (Fig. 2c), confirming once more that we are in the SC-regime.

Electrically, both reference device on ITO and SC-devices have an almost identical $V_{OC}$ of 1.151 and 1.146 V, respectively (Fig. 1c, and Table 1). Thus, with the lowering of the $E_{opt}$, the $E_{opt}-qV_{OC}$ losses are reduced from 0.576 V for the reference device, to only 0.525 V for the SC-device with the most redshifted LP peak ($d = 55$ nm). This value for $E_{opt}-qV_{OC}$ losses is exceptionally low for OSCs, and comes within the range of other efficient photovoltaic technologies[30].

Overall, the utilization of the 25-nm Ag front electrode needed to support SC in the SC-devices, induces an increased reflection in the spectral range of SubNc and Cl$_6$-PhOSubPc exciton absorption, resulting in a lowered short-circuit current density ($J_{SC}$). The fill factors of the SC and reference device are similar, FF = 50% and FF = 53.5%, respectively. In total, SC reduces the $E_{opt}-qV_{OC}$ energy losses, but has not been found to optimize the efficiency of the studied single-junction OSCs. However, most of the photon flux which is not converted to photocurrent is reflected by the SC device, and therefore not lost for photovoltaic harvesting. For example, we predict that SC can improve the performance of multi-junction devices by steepening the absorption edge of the subcell with the lowest $E_{opt}$, hereby harvesting more photons in the spectral region around $E_{opt}$, while the remaining subcells absorb the reflected light. This would enable a reduction in voltage losses while simultaneously increasing photon harvesting.

**Reducing the driving force for charge transfer with strong coupling.** The fact that $V_{OC}$ remains unaffected warrants investigation of the CT-state energetics when embedding the photoactive layers into resonant optical cavities. Thus, we fabricated SC-devices employing C$_{60}$ as acceptor instead of Cl$_6$-PhOSubPc. The use of C$_{60}$, having a deeper lowest unoccupied molecular orbital (LUMO), results in a more pronounced CT absorption and emission feature in sensitively measured EQE and EL spectra,

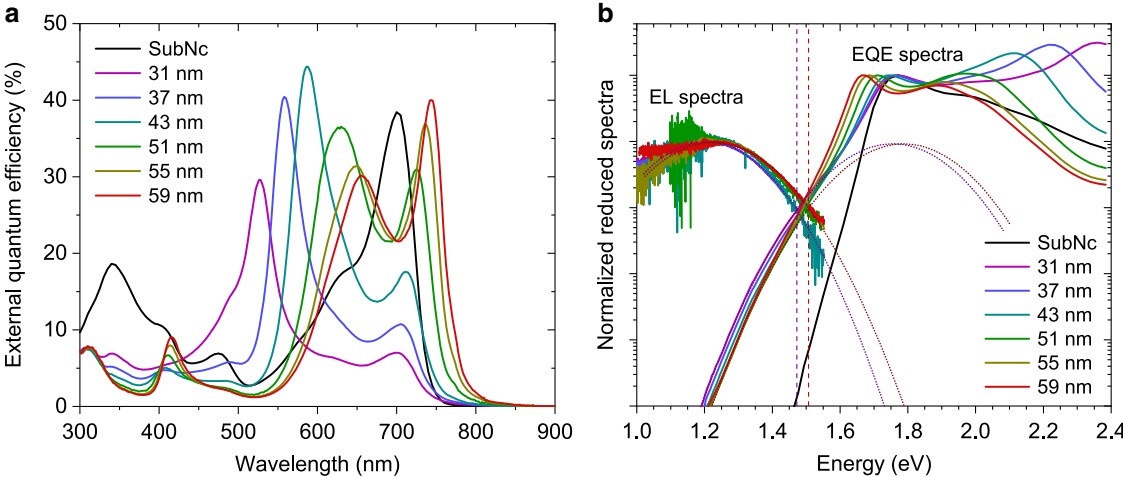

**Fig. 3** Effects of strong coupling on EQE and electroluminescence of SubNc/$C_{60}$ based cells. **a** EQE spectra for strongly coupled SubNc/$C_{60}$ solar cells, for varying transport layer thickness $d$. EQE of a reference solar cell utilizing only SubNc ('SubNc', black) is included for comparison. In the strongly coupled devices, the SubNc peak is split into two polariton peaks which redshift for increasing $d$. **b** Sensitively measured normalized reduced EQE and EL spectra of the same devices. The low-energy EQE and high-energy EL edges of the most blue-shifted and red-shifted spectra are fitted (purple and red dashed lines, respectively) and the crossing point of their Gaussian lineshapes (purple and red dotted lines) provide the CT-state energy in each case

respectively. This is in contrast to the SubNc/$Cl_6$-PhOSubPc devices which exhibit minimal driving force, and any CT-state-related features are absent in their EQE and EL spectra (Supplementary Fig. 6).

Six SubNc/$C_{60}$ devices are investigated with different transport layer thicknesses $d$, ranging from 31 to 59 nm. Figure 3a shows the EQE spectra of the devices, where the SubNc peak is split into two polariton branches. As compared to the EQE spectrum of a solar cell utilizing only SubNc, peak splitting and polariton formation is observed. Again, the LP branch redshifts for increasing $d$ from 700 nm (1.771 eV) to 743 nm (1.669 eV), thus reducing the $E_{opt}$ by 102 meV. Looking closer at the subgap region (Fig. 3b), a red-shifted feature related to SubNc/$C_{60}$ CT-states is observed. Additionally, the EL spectra of the SubNc/$C_{60}$-based SC-devices are dominated by CT-state emission with a signal centered at around 1.240 eV (Fig. 3b), significantly red-shifted as compared to the LP emission which dominates the EL spectrum of SubNc/$Cl_6$-PhOSubPc devices (Fig. 2a).

Taking into account the two extreme cases (most blue-shifted and red-shifted), we find that $E_{CT}$ changes only marginally by 20–30 meV when varying the transport layer thickness $d$ (Supplementary Fig. 7). This shift is rather small as compared to the 102 meV red-shift of the LP EQE peak over the same thickness range (Fig. 3b). Thus, we conclude that although the singlet excited state energetics of the SC-devices alter significantly, $E_{CT}$ remains rather unaffected by SC. This is confirmed by an alternative method to determine interfacial energetics, using temperature-dependent $V_{OC}$ measurements. $V_{OC}$ values extrapolated to 0 K ($V0$) have been shown to correspond to $E_{CT}$ values extrapolated to 0 K[31]. For all cavity enhanced SubNc/$C_{60}$ devices $V_{OC}$'s extrapolated to 0 K are indeed found to be similar (Supplementary Fig. 8). Since SC requires strong absorption, the CT-state absorption is generally too weak to support strong coupling between the cavity photon and the CT-excitons. Interestingly enough, the reduction of $E_{opt}$ together with the constant $E_{CT}$ imply that the driving force for electron transfer ($E_{opt}$–$E_{CT}$) in these devices is reduced by increasing the cavity length. Indeed, we observe a reduction of the driving force by 130 meV in these devices, from 299 meV ($d = 31$ nm) to 159 meV ($d = 59$ nm).

For all the SubNc/$C_{60}$ devices, $V_{OC}$ remains rather constant at around 0.80 V and, therefore, the $E_{opt}$–$qV_{OC}$ losses decrease from

0.980 V down to 0.874 V over the investigated thickness $d$ range (Supplementary Table 1 and Fig. 4). $V_{rad}$ varies minimally, and since $V_{OC}$ and $E_{CT}$ remain also constant, the radiative losses $\Delta V_{rad} = E_{CT}/q - V_{rad}$ increase by 45 mV and $\Delta V_{nonrad} = V_{rad} - V_{OC}$ remain rather constant around 0.380 V. In total, we find that the recombination losses ($\Delta V_{rad}$ and $\Delta V_{nonrad}$) are not affected drastically by SC. This implies that the main contribution of SC in SubNc/$C_{60}$ devices is the reduction of the total photon energy losses ($E_{opt}$–$qV_{OC}$), by reducing the driving force for electron transfer from SubNc to $C_{60}$. Note that this decrease in driving force does not go at the cost of a reduced peak EQE value.

**Steepening the absorption edge with strong coupling**. As discussed above, also for the SubNc/$Cl_6$-PhOSubPc devices, where the CT state is shifted in energy close to the SubNc exciton, SC can significantly decrease the voltage losses. Typically, disordered materials, such as organic semiconductors exhibit shallow absorption edges with increased absorption of photons with energy below the material's $E_{opt}$, but decreased absorption of photons at $E_{opt}$. Below, we demonstrate that the absorption edges of the SubNc/$Cl_6$-PhOSubPc device can be steepened to values comparable to crystalline inorganic semiconductors by SC, hereby reducing the energy losses for strongly absorbed photons (Fig. 5a). In order to evaluate the steepness of the absorption edge, the so-called Urbach energy ($E_U$) is determined by fitting the slope of the exponential part of the low-energy EQE tail[32]:

$$a(E) \propto \text{EQE}(E) \propto \exp\left(\frac{E}{E_U}\right) \qquad (2)$$

$E_U$ is typically extracted from sensitive absorption spectra obtained via photothermal deflection spectroscopy, however, sensitively measured EQE spectra can be also employed[33,34], since the internal quantum efficiency is constant in the low-energy tail region[35]. For the investigated SC-devices, $E_U$ decreases from 22.4 meV for the reference SubNc/$Cl_6$-PhOSubPc device ('SubNc' in Fig. 4b, c) down to 15.6 meV for the SC-device with $d = 55$ nm ('SC-SubNc' in Fig. 4b, c). It is worth noting that since $E_U$ is less than $k_B T$ at room temperature, for both reference and SC-devices, a significant increase in $V_{OC}$ is not expected due to the absorption edge steepening (see also Supplementary Note 2). Figure 4b, adapted from De Wolf et al.[33] and Jean et al.[36], compares the $E_U$

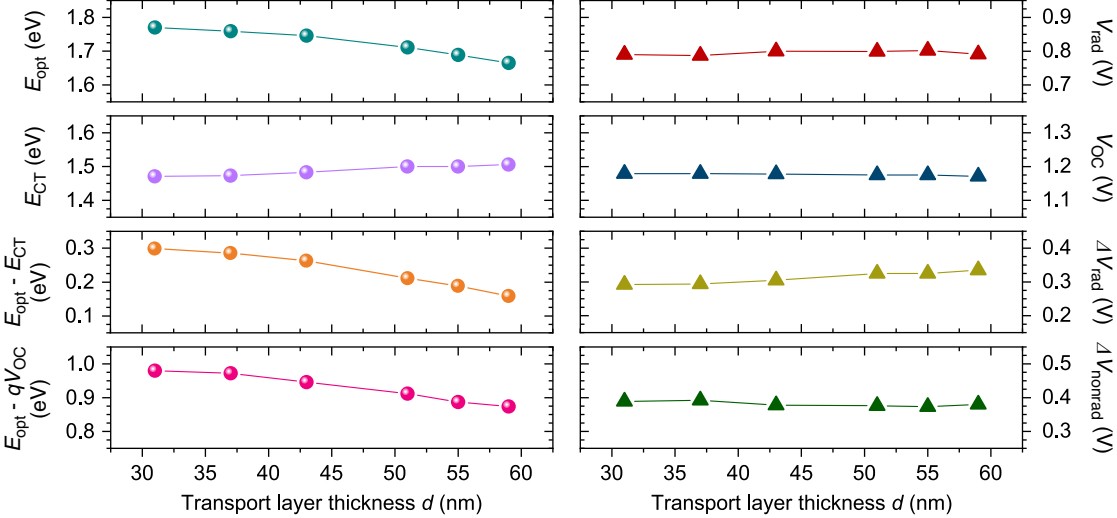

**Fig. 4** Voltage and energy losses of SubNc/$C_{60}$ SC-devices with varying transport layer thickness $d$. The $E_{opt}$ of the devices corresponds to the peak of the LP branch $\lambda_{peak}$ (Fig. 3a), and $E_{CT}$ is determined from the crossing point between appropriately normalized reduced EQE and EL spectra (Fig. 3b), which is found to be approximately the same for the investigated devices. With increasing transport layer thickness, the driving force ($E_{opt}-E_{CT}$) and the total energy losses ($E_{opt}-qV_{OC}$) decrease. $\Delta V_{rad}$ and $\Delta V_{nonrad}$ correspond to the voltage losses related to radiative and non-radiative recombination, respectively, and remain rather unaffected by strong coupling

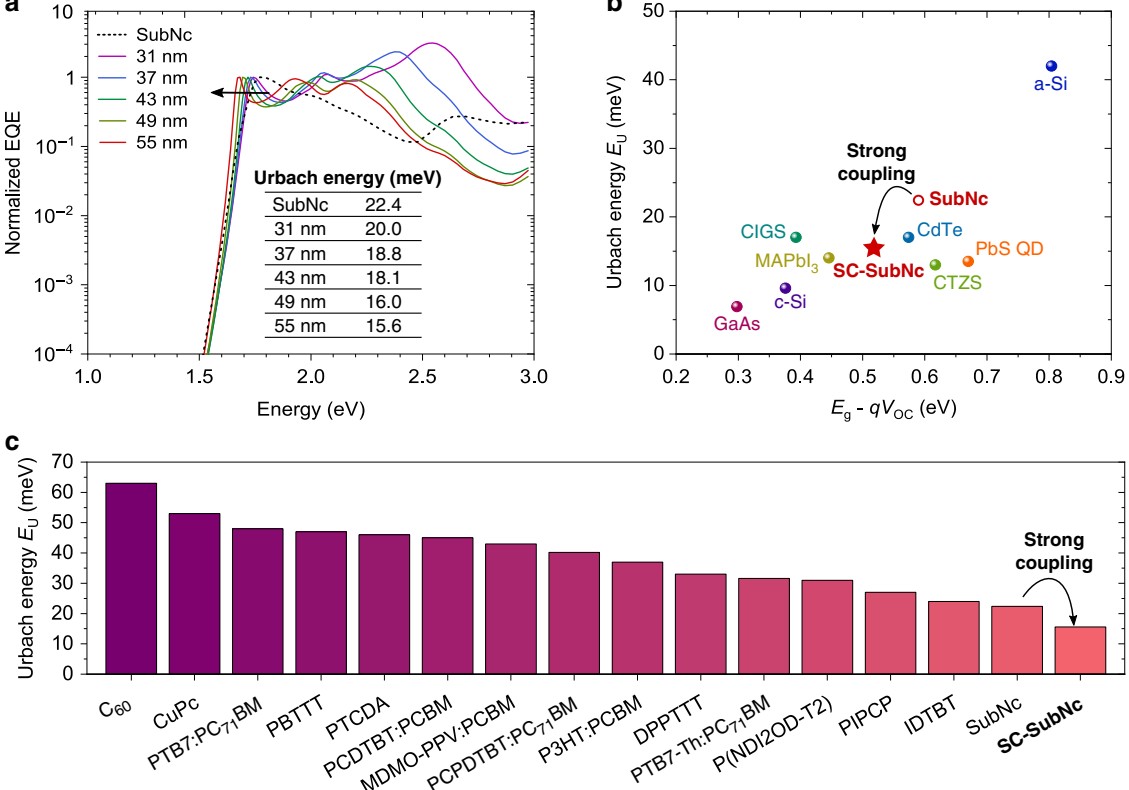

**Fig. 5** Steepening the absorption edge for reduced photon energy losses with strong coupling. **a** Sensitively measured EQE spectra of the investigated SubNc/$Cl_6$-PhOSubPc based SC-devices for various transport layer thicknesses $d$, including the reference solar cell employing only SubNc ('SubNc'). Increasing $d$ red-shifts the device absorption, steepening the absorption edge. **b** Urbach energy ($E_U$) versus $E_g-qV_{OC}$ losses for the SubNc/$Cl_6$-PhOSubPc based SC-device ('SC-SubNc'), exhibiting the $E_U = 15.6$ meV and energy losses of 0.525 eV, and record solar cells of other inorganic photovoltaic technologies. The reference SubNc/$Cl_6$-PhOSubPc device ('SubNc') is also included to demonstrate how strong coupling reduces both $E_U$ and photon energy losses. Adapted by De Wolf et al. [33] and Jean et al. [36]. **c** $E_U$ values for various organic photovoltaic materials and blends. The value of 15.6 meV for the 'SC-SubNc' device is the lowest for organic materials. Values of materials and blends, which were not investigated in this work, were taken from literature[34,37-40]

values for our SC-devices with solar cells of various photovoltaic technologies versus the $E_g$–$qV_{OC}$ losses in each case. A clear linear relationship is observed, confirming that a steep absorption edge is generally accompanied by low photon energy losses in photovoltaic devices. Furthermore, the $E_U$ = 15.6 meV obtained for the SC-device is the lowest for organic materials (Fig. 4c), and comparable to those of highly efficient photovoltaic materials, such as lead-based perovskites (MAPbI$_3$), cadmium tellurium (CdTe), and copper indium gallium selenide (CIGS), among others (Fig. 4b)[34,37–40].

## Discussion

Our work shows that strong coupling allows to optically manipulate material properties of organic semiconductors which are classically determined by molecular properties. This approach opens unexplored possibilities of application in organic optoelectronic devices. For OSCs, the formation of polaritons can be used to modify the device's absorption onset and tune the device's $E_{opt}$. At the same time, the $V_{OC}$ in those devices remains unaffected, making it possible to reduce the $E_{opt}$–$qV_{OC}$ losses by 60–100 meV. For SubNc/C$_{60}$-based devices, with a considerably large driving force for electron transfer, we find that the energetics of CT-states are not affected by the cavity device architecture, and we attribute this to the weak CT-absorption, which cannot support strong coupling. Since $E_{CT}$ remains unaffected, the reduction of $E_{opt}$ results in a lowering of the driving force for electron transfer, purely induced by SC with the SubNc exciton. In the SubNc/Cl$_6$-PhOSubPc devices, with an already minimized driving force for electron transfer, strong coupling steepens the absorption edge of OSCs. In those devices, Urbach energy as low as 15.6 meV was achieved, being comparable to that of crystalline inorganic semiconductors. The combination of reduced driving force and steepened absorption edge results in a SC SubNc/Cl$_6$-PhOSubPc device with overall $E_{opt}$–$qV_{OC}$ energy losses for strongly absorbed photons at 0.525 eV. This value is comparable with record cells of other efficient photovoltaic technologies.

## Methods

**Device fabrication.** The solar cells shown in this publication are processed by thermal evaporation in a custom made vacuum system (Kurt J. Lesker, USA) with a base pressure of $10^{-8}$ mbar. During a processing run, different masks and movable shutters enable the variation of the device stacks or processing parameters, offering the possibility to produce and compare various devices at the same processing conditions. Each device is fabricated onto either clean glass substrates or substrates with pre-structured ITO (Thin Film Devices, USA), which undergo an ozone treatment before being transferred into the vacuum chamber. Every investigated device is bottom illuminated, employing either thin Ag or ITO as anode and a 100-nm-thick Ag cathode. The area of the devices was 6.44 mm$^2$, defined as the overlap between anode and the Ag cathode. All the used materials are purified twice in-house by vacuum gradient sublimation. The solar cells are encapsulated in nitrogen atmosphere with a transparent encapsulation glass, fixed by UV-hardened epoxy glue.

**Device structures.** BF-DPB (Synthon, Germany) p-doped by NDP9 (Novaled, Germany), at a 2 wt% mixing ratio, are deposited on top of the anode and function as doped hole transporting layer. An intrinsic BF-DPB layer of 5 nm thickness is used as exciton blocking layer, avoiding also the contact between the active layer and the dopant. In the active layer, SubNc (Lumtec, Taiwan) is used as donor and C$_{60}$ (Creaphys, Germany) or Cl$_6$-PhOSubPc (Lumtec, Taiwan) are used as acceptors. Donor and acceptor layers are sequentially deposited forming a planar heterojunction. In the reference devices, 8 nm of BPhen (ABCR, Germany) are used as exciton blocking and electron transporting layer. In the strong coupling devices, 5 nm of BPhen are used as exciton blocking layer followed by a layer of Bis-Hfl-NTCDI (synthesized in-house) n-doped by W$_2$(hpp)$_4$ (Novaled, Germany), at a 7 wt% mixing ratio, acting as ETL. As metal top contact, 100 nm of Ag are used. Each organic material is evaporated at a rate of 0.3 Å/s, apart from SubNc and Cl$_6$-PhOSubPc which are deposited at 0.5 Å/s. The Ag electrodes are deposited at 1 Å/s. All layer thicknesses are monitored with calibrated quartz crystal microbalances (QCM). In summary, the device structures of the solar cells shown in this work are the following (the number in parentheses denote the nominal layer thickness in

nm); *reference SubNc/Cl$_6$-PhOSubPc device:* ITO/BF-DPB:NDP9(30)/BF-DPB(5)/SubNc(12)/Cl$_6$-PhOSubPc(20)/BPhen(8)/Ag(100); *SubNc/Cl$_6$-PhOSubPc SC-devices:* Ag (25)/BF-DPB:NDP9($d$)/BF-DPB(5)/SubNc(12)/Cl$_6$-PhOSubPc(12)/BPhen(5)/Bis-Hfl-NTCDI:W$_2$(hpp)$_4$($d$)/Ag(100); *SubNc/C$_{60}$ SC-devices:* Ag (25)/BF-DPB:NDP9($d$)/BF-DPB(5)/SubNc(12)/C$_{60}$ (12)/BPhen(5)/Bis-Hfl-NTCDI:W$_2$(hpp)$_4$($d$)/Ag(100); *SubNc-only device* (shown in Fig. 3b): ITO/BF-DPB:NDP9 (30, 7 wt%)/BF-DPB(5)/SubNc(12)/BPhen(8)/Ag(100)

**Current voltage (j–V) measurements.** $j$–$V$ measurements are carried out in ambient conditions using a source measurement unit (SMU 2400 Keithley, USA) and a simulated AM1.5 G illumination (16S-003-300-AM1.5 G sunlight simulator, Solar Light Co., USA). A silicon photodiode (Hamamatsu S1337) is used as reference. Spectral mismatch is taken into account during the measurement.

**EQE measurements.** EQE measurements are performed using a xenon lamp (Oriel Xe Arch-lamp Apex, Newport, USA), a monochromator (Cornerstone 260 1/4m, Newport, USA), an optical chopper, and a lock-in amplifier (SR 7265, Signal Recovery, USA). The EQE of the OSCs is measured with an aperture mask (2.78 mm$^2$) and without bias light. A silicon photodiode (Hamamatsu S1337, JP) is used as reference. For the angle-resolved EQE measurements a custom goniometer base is used additionally.

**Sensitive EQE measurements.** The light of a quartz halogen lamp (50 W) is chopped at 140 Hz and coupled into a monochromator (Newport Cornerstone 260 1/4m, USA). The resulting monochromatic light is focused onto the OSC, its current at short-circuit conditions is fed to a current pre-amplifier before it is analyzed with a lock-in amplifier (Signal Recovery 7280 DSP, USA). The time constant of the lock-in amplifier is chosen to be 500 ms and the amplification of the pre-amplifier is increased to resolve low photocurrents. The EQE is determined by dividing the photocurrent of the OSC by the flux of incoming photons, which is obtained with calibrated silicon (Si) and indium–gallium–arsenide (InGaAs) photodiodes.

**Temperature-dependent $V_{OC}$ measurements.** A Keithley SMU2635A is used to control the light intensity of three white LEDs in series (APG2C3-NW, Roithner, Austria) used as light source for our OSCs. A Keithley dual channel SMU2602A measures both the $V_{oc}$ and the illumination intensity with a S2387-66R Si Photodiode (Japan). To vary the cell temperature, the devices are placed in vacuum on a copper block, which is connected to a Peltier element from Peltron GmbH (Germany) controlled by a BelektroniG HAT Control device (Germany). The measurement equipment is controlled with the software SweepMe! (https://sweep-me.net).

**EL measurements.** EL spectra are acquired with an Andor SR393i-B spectrometer equipped with an iDus silicon (DU420A-BR-DD) and an InGaAs (DU491A-1.7) detector array. The spectral response of the setup (detector and grating) is evaluated by means of a calibrated lamp (Oriel 63355). The EL spectra are acquired by driving the solar cells with a Keithley 2400 SMU at injection currents equivalent to their short-circuit current at 1 sun.

**Reporting summary.** Further information on research design is available in the Nature Research Reporting Summary linked to this article.

## Data availability
The datasets supporting this publication can be accessed via the PURE repository at https://doi.org/10.17630/c8bb75db-b18b-4eab-b609-ac4628094554

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

## Acknowledgements

This work was supported by the German Federal Ministry of Education and Research (BMBF) through the Innoprofile project "Organische p-i-n Bauelemente2.2" (FKZ 03IPT602X). M.C.G. acknowledges funding from the Volkswagen Foundation (no. 93404). A.M. acknowledges funding through an individual fellowship of the Deutsche Forschungsgemeinschaft (404587082). J.K. thanks the German Academic Exchange Service (DAAD) for financial support (Grant no. 57214224). U.H. and D.N. acknowledge funding from the German Research Foundation (DFG) within the collaborative research center 951 "Hybrid Inorganic/Organic Systems for Opto-Electronics (HIOS).

## Author contributions

V.C.N., A.M., B.S., D.S. and K.V. designed the experiments. V.C.N. optimized the photovoltaic devices and performed the j–V, EQE, and angle-resolved EQE measurements. A.M. carried out the simulations and calculations of the polariton-based solar cells. J.K. measured the sensitive EQE spectra. X.J. performed the temperature-dependent $V_{OC}$ measurements. V.C.N. and U.H. measured the EL spectra. V.C.N. and J.B. performed the voltage loss analysis. D.N. and M.C.G. supervised their team members involved in this project. K.V. supervised the overall project. All authors contributed to analysis and writing.

## Additional information

**Competing interests:** The authors declare no competing interests.

