## [Peer Review File · Nature Communications]

Reviewers' comments:

Reviewer #1 (Remarks to the Author):

In this work the authors, by emending the device in an optical microcavity, investigated the effect of light-matter coupling on the organic solar cell properties. The polaritonic effect leads to a splitting of the lowest optical transition of the donor and an overall red-shift of the absorption band. However, the increase in the light-matter coupling does not have any effect on the open circuit voltage of the cell. At the same time the replacement of the ITO with Ag electrode reduces the device EQE and subsequently the current. So, overall the paper shows that the use of an optical cavity does not provide any benefit for solar cells.

I find nevertheless the study very interesting and recommend publication given that the authors will consider making the following modifications to the paper.

(i) One should use the same reference for all devices to describe the voltage losses. In this case this would be the absorption of the pristine donor. I understand that the authors try to put a positive spin on their findings, but It is unreasonable and misleading to claim that the light-matter coupling reduces the voltage losses since voltage is not affected at all by this interaction.

(ii) The steepness of the absorption edge is interpreted in terms of the Urbach energy (E_u). At this point, it looks that the change in the absorption steepness due to the light-matter coupling is accompanied with no effect on the voltage. It might be useful to stress this point and to consider an alternative interpretation of the absorption edge steepness.

Reviewer #2 (Remarks to the Author):

Nikolis et al. present a study on organic solar cells where they employ strong light-matter coupling and demonstrate a reduced offset between the minimum absorbed photon energy in the donor material and the obtained open-circuit voltage. By fabricating an optical cavity the authors find strong light-matter coupling with a resulting red-shift in the absorption of the photoactive layer along with a steepening of the absorption edge. The corresponding Urbach energy is shown to decrease upon strengthening of the light-matter coupling.

The work is very appealing and the paper is well written, concise and clear. To my point of view it is therefore in principle suitable for publication in Nature Communication. However, I would suggest that the authors include some further analysis and experiments to complement their study as outlined in the following:

1) The first point I would like to raise is that the SC device in Table 1 has a lower J_{sc} value. As well explained by the authors this is to be expected as a consequence of the increased reflection of the "transparent" electrode when going from ITO to the 25 nm thick Ag layer. However, a quick estimate shows that this reduced generation should lead to a roughly 12 mV lower V_{oc} for the SC device (when assuming about 60 mV per decade of light intensity). Do the authors have any explanation for the fact that the V_{oc} values are identical?

2) The authors state that CT state energies do not change upon strengthening of the optical cavity

effects. I think measuring the Voc as a function of temperature for the SC and the reference device to see whether they coincide with their effective bandgaps obtained by extrapolating the experimental Voc(T) data to T=0K would be a very interesting amendment of the work presented thus far.

Minor point:

Under Experimental methods, the authors write that the electroluminescence measurements were carried out at injection currents equivalent to the short-circuit currents of the solar cells. I guess they refer to "1 sun"? Please add that information.

Best regards
Uli Würfel

Reviewer #3 (Remarks to the Author):

In their manuscript, Nikolis et al. investigate organic solar cells in the strong light-matter interaction regime. They observe polariton branches that give rise to a steepened absorption edge while the charge-transfer state remains unaffected, leading to an effective reduction of energy loss. As the authors discuss candidly, using resonant structures lowers the power conversion efficiency but nevertheless might be interesting for homo-tandem / multi-junction cells. This work is a nice example that even without ultra-strong light-matter coupling the energy levels can be modified in a useful way, and it provides a novel angle for organic photovoltaics.

The paper is very well written, and the arguments and conclusions are sound and convincing. It includes a comprehensive set of data that clearly shows the effects of the strong coupling regime, with all dependencies that are of interest. I have only some minor comments that should be addressed before publication:

- 1) The present device area is only a few mm². Can the concept be realistically expanded to larger devices, given that already small changes in thickness or homogeneity can lead to a pronounced shift of the resonance wavelength of the cavity?
- 2) Can the authors explain the additional modulation as a function of angle in Fig. S4 top right?
- 3) The real Stokes shift can be approximately obtained from the peak centers of absorption and PL peaks only for thin layers without significant reabsorption. Is this condition met for the data in Fig. S1? The Stokes shift should also be given in meV (not only nm).
- 4) Line 340: Should be 0.52 eV instead of 0.52 meV.
- 5) Line 392: The bottom Ag(25) layer is missing in the description of the SC layer stack.

Reviewers' comments:

Reviewer #1 (Remarks to the Author):

In this work the authors, by emending the device in an optical microcavity, investigated the effect of light-matter coupling on the organic solar cell properties. The polaritonic effect leads to a splitting of the lowest optical transition of the donor and an overall red-shift of the absorption band. However, the increase in the light-matter coupling does not have any effect on the open circuit voltage of the cell. At the same time the replacement of the ITO with Ag electrode reduces the device EQE and subsequently the current. So, overall the paper shows that the use of an optical cavity does not provide any benefit for solar cells.

I find nevertheless the study very interesting and recommend publication given that the authors will consider making the following modifications to the paper.

Authors' answer to Reviewer #1:

Thank you for your positive assessment. Despite the reduction of the $E_{opt} - V_{OC}$ losses, the steepening of the absorption edge, as well as the reduction of the driving force, the strong coupling effects do indeed not improve the efficiency of single-junction devices. This is due to the moderate transparency of the utilized contacts in the strongly absorbing spectral region, which considerably reduces the total photocurrent. However, we want to emphasize that this problem can be overcome: Most of the light which is not converted to photocurrent is reflected with minor parasitic absorption and, therefore, not lost but can be harvested by a second solar cell in a (parallel connected or four terminal) tandem configuration. This was appreciated by Reviewer #3.

We have strengthened this aspect in the manuscript as follows: (updated text in the manuscript is highlighted in yellow)

"In total, SC reduces the $E_{opt} - qV_{OC}$ energy losses, but has not been found to optimize the efficiency of the studied single-junction OSCs. However, most of the photon flux which is not converted to photocurrent is reflected by the SC device, and therefore not lost for photovoltaic harvesting. For example, we predict that SC can improve the performance of multi-junction devices by steepening the absorption edge of the subcell with the lowest E_{opt} , hereby harvesting more photons in the spectral region around E_{opt} , while the remaining subcell absorb the reflected light. This would enable a reduction in voltage losses while simultaneously increasing photon harvesting." (Lines 217-224)

(i) One should use the same reference for all devices to describe the voltage losses. In this case this would be the absorption of the pristine donor. I understand that the authors try to put a positive spin on their findings, but It is unreasonable and misleading to claim that the light-matter coupling reduces the voltage losses since voltage is not affected at all by this interaction.

Authors' answer to Reviewer #1:

“ $E_{opt}-qV_{OC}$ ” represents a lower limit to the energy losses in organic solar cells. It compares the maximum potential provided by the cell to the minimum energy of photons absorbed by the device. Obtaining V_{OC} is rather straightforward and based on basic PV characterization. However, the appropriate determination of E_{opt} in organic materials and devices has been the topic of vivid discussion in the OPV community the last years.^{1,2}

In the case of strong coupling the use of the E_{opt} of the pristine donor as universal reference for all the devices is in our opinion not justified. In the manuscript, we prove that strong light matter coupling occurs, which implies that in this case, it's not the donor which determines the optical gap, but the lower polariton branch of the light-matter state. Using strong coupling, actual matter properties are affected and thus it is correct to compare V_{OC} to the “new” strong coupling material inside the device.

While V_{OC} is indeed not significantly affected by SC, our work shows a reduction of energy losses for photons absorbed by the lower polariton branch. Up to now, E_{opt} has been considered as a material property which could not be altered significantly. The essence of our work is that we demonstrate that $E_{opt}-qV_{OC}$ offset can be reduced by bringing the device's E_{opt} closer to V_{OC} , by the realization of a new polariton state with energy below the exciton optical gap. Hence, our statement that strong coupling can be used to reduce the overall photon energy losses is definitely reasonable. To clarify that a new state with energy below the exciton energy is created but also the fact that V_{OC} remains almost unchanged, we added the following sentence to the main text:

“In this work, we explore a strategy to reduce photon energy losses in OSCs, through the use of strong light-matter coupling (SC). Hereby, we induce new states which exhibit a red-shifted E_{opt} as compared to the pristine absorbers. The crucial point is that we benefit from an unchanged V_{OC} , by bringing the device's E_{opt} closer to V_{OC} ” (Lines 64-67)

The fact that V_{OC} is not lowered, even though there clearly is a redshift in the device's absorption, is further explored by a detailed voltage analysis which shows that recombination losses are not affected significantly by SC.

This is stated in the main text:

“In total, we find that the recombination losses (ΔV_{rad} and ΔV_{nonrad}) are not affected drastically by SC. This implies that the main contribution of SC in SubNc/C₆₀ devices is the reduction of the total photon energy losses ($E_{opt} - qV_{OC}$), by reducing the driving force for electron transfer from SubNc to C₆₀.”

To make this point clearer, we updated the Figure 4 of the main text, including all the voltage losses due to recombination (ΔV_{rad} and ΔV_{nonrad}) together with overall photon energy losses ($E_{opt} - qV_{OC}$) and driving force ($E_{opt} - E_{CT}$). There it is clear that SC reduces significantly only the energy losses $E_{opt} - qV_{OC}$ and $E_{opt} - E_{CT}$.

Moreover, in the caption of Figure 4 is now written: (updated text is highlighted in yellow)

“Figure 4. Voltage and energy losses of strongly coupled SubNc/C₆₀ devices with various transport layer thicknesses *d*. The E_{opt} of the devices corresponds to the peak of the LP branch λ_{peak} (Figure 3a), and E_{CT} is determined from the crossing point between appropriately normalized reduced EQE and EL spectra (Figure 3b), which is found to be approximately the same for the investigated devices. With increasing transport layer thickness, the driving force ($E_{opt} - E_{CT}$) and the total energy losses ($E_{opt} - qV_{OC}$) decrease. ΔV_{rad} and ΔV_{nonrad} correspond to the voltage losses related to radiative and non-radiative recombination, respectively, and remain rather unaffected by strong coupling.” (Lines 289-292)

1. Vandewal, K., Benduhn, J. & Nikolis, V. C. How to determine optical gaps and voltage losses in organic photovoltaic materials. *Sustainable Energy & Fuels* 2, 538–544 (2018).
2. Wang, Y. et al. Optical Gaps of Organic Solar Cells as a Reference for Comparing Voltage Losses. *Advanced Energy Materials* 8, 1801352 (2018).

(ii) The steepness of the absorption edge is interpreted in terms of the Urbach energy (E_u). At this point, it looks that the change in the absorption steepness due to the light-matter coupling is accompanied with no effect on the voltage. It might be useful to stress this point and to consider an alternative interpretation of the absorption edge steepness.

Answer to Reviewer #1:

We thank the Reviewer for giving us the opportunity to clarify this point.

It is true that, in our experiments, the change in the absorption steepness due to strong coupling does not have a significant effect on the V_{OC} of the investigated devices. However,

we feel that an alternative interpretation of the absorption steepness is currently not needed to explain this effect, for the following reason:

The Urbach energy (E_U) represents the slope of the broadened absorption tail of disordered semiconductors. In reference 35 of the main text of the revised manuscript, Jean et al. have derived the detailed balanced efficiency limit of solar cells with pronounced band tailing, as a function of their Urbach edge.¹

They find a threshold E_U value at $k_B T$, where two regimes are observed:¹

1. for $E_U > k_B T$, the radiative limit to V_{OC} , V_{rad} decreases rapidly. A reduction of E_U at this regime would lead to a significantly increased V_{OC} .
2. for $E_U < k_B T$, V_{rad} increases only very slightly due to the slight increase in J_{SC} .

Our measurements were performed at room temperature ($T = 298$ K), where $k_B T$ is equal to 25.8 meV. For our reference device, E_U is already at 22.4 meV, since SubNc exhibits in general a very steep absorption edge. By employing strong coupling, we reduce E_U to 15.6 meV in the best case. Thus, it is clear that the whole E_U optimization by SC occurs in the " $E_U < k_B T$ " regime for our samples, where V_{OC} is not expected to depend strongly on E_U . Based on the model of Jean et al. for disordered semiconductors¹, we estimate that the reduction of E_U from 22.4 meV to 15.6 meV should lead to a voltage increase of approximately 40 mV. Our calculations for the V_{rad} of the investigated devices lead to a 25 mV increase (see Table S2), which is in the same range.

Now, for the real V_{OC} of our devices we have to consider losses due to non-radiative losses, as well as optical losses (due to parasitic absorption and reflection), reducing J_{SC} . These can dissipate this predicted marginal gain in voltage and lead to a seemingly non-optimized photovoltage.

To make this point clear in our manuscript, we added the following sentence: (changes are highlighted in yellow)

"For the investigated SC-devices, E_U decreases from 22.4 meV for the reference SubNc/Cl₆-PhOSubPc device ("SubNc" in Figures 4b and 4c) down to 15.6 meV for the SC-device with $d = 55$ nm ("SC-SubNc" in Figure 4b and 4c). It is worth noting that since E_U is less than $k_B T$ at room temperature, for both reference and SC-devices, a significant increase in V_{OC} is not expected due to the absorption edge steepening (see also Supplementary Information section 10)." (Lines 310-313)

Moreover, we added an extended reply to the Reviewer's comment as Section 10, in the revised Supplementary Information.

1. Jean, J. et al. Radiative Efficiency Limit with Band Tailing Exceeds 30% for Quantum Dot Solar Cells. *ACS Energy Lett.* **2**, 2616–2624 (2017).

Reviewer #2 (Remarks to the Author):

Nikolis et al. present a study on organic solar cells where they employ strong light-matter coupling and demonstrate a reduced offset between the minimum absorbed photon energy in the donor material and the obtained open-circuit voltage. By fabricating an optical cavity the authors find strong light-matter coupling with a resulting red-shift in the absorption of the photoactive layer along with a steepening of the absorption edge. The corresponding Urbach energy is shown to decrease upon strengthening of the light-matter coupling.

The work is very appealing and the paper is well written, concise and clear. To my point of view it is therefore in principle suitable for publication in Nature Communication. However, I would suggest that the authors include some further analysis and experiments to complement their study as outlined in the following:

1) The first point I would like to raise is that the SC device in Table 1 has a lower J_{sc} value. As well explained by the authors this is to be expected as a consequence of the increased reflection of the "transparent" electrode when going from ITO to the 25 nm thick Ag layer. However, a quick estimate shows that this reduced generation should lead to a roughly 12 mV lower V_{oc} for the SC device (when assuming about 60 mV per decade of light intensity). Do the authors have any explanation for the fact that the V_{oc} values are identical?

Reply to Reviewer #2:

We fully agree with the perspective of Reviewer #2. Even looking at the Figure 1c it is clear that the two jV curves have not identical V_{oc} 's. The values were rounded to two decimals, but the exact values obtained by our measurement setup correspond to 1.146 V for the SC-device, and 1.151 V for the reference, therefore we obtain an offset of 5 mV. This is 7 mV less than the predicted 12 mV difference, which could be due to the fact that much thicker transport layers have been used for the SC-devices, or even due to measurement artefacts, as V_{oc} differences within 10 mV are very hard to distinguish.

Nevertheless, in order to clarify this point, we updated the main text as follows: (changes are highlighted in yellow)

- All the voltage or energy values, as well as every other value calculated using these quantities are now displayed with three decimals throughout the whole main text and SI.
- the sentence "Electrically, both reference device on ITO and SC-devices have a V_{oc} of 1.15 V." is now written as "Electrically, both reference device on ITO and SC-devices have an almost identical V_{oc} of 1.151 V and 1.146 V, respectively" (Lines 197-198)

2) The authors state that CT state energies do not change upon strengthening of the optical cavity effects. I think measuring the V_{oc} as a function of temperature for the SC and the

reference device to see whether they coincide with their effective bandgaps obtained by extrapolating the experimental $V_{OC}(T)$ data to $T=0K$ would be a very interesting amendment of the work presented thus far.

Reply to Reviewer #2:

Temperature-dependent V_{OC} measurements are indeed an important confirmation of our results, and we thank the reviewer for this suggestion. We complemented our EQE and EL measurements of SubNc/C₆₀ SC-devices with T-dependent V_{OC} measurements, and the results show that the V_{OC} 's ($T=0K$) of the respective samples are very similar, exhibiting a variation of 42 mV, being within the same range of variation as E_{CT} obtained via EQE and EL measurements (35 meV). It is noteworthy that, in absolute numbers, the obtained V_{OC} 's ($T=0K$) values are lower than the E_{CT} values obtained via EQE and EL measurements. This is, however, expected since the EQE and EL measurements were conducted at room temperature (RT), and E_{CT} is known to exhibit some temperature dependence.¹

For this set of measurements, we created a new section in the Supplementary Information, where both the graph and table are shown and the results are discussed. The measurements were conducted at a light intensity approximately equal to 1 sun. A new section was added to the Experimental Methods, providing the necessary information about our T-dependent V_{OC} measurements.

Moreover, in the main text is now written: (changes are highlighted in yellow)

"This is confirmed by an alternative method to determine interfacial energetics, using temperature-dependent V_{OC} measurements. V_{OC} values extrapolated to 0 K (V_0) have been shown to correspond to E_{CT} values extrapolated to 0 K. [reference: Vandewal, K., Tvingstedt, K., Gadisa, A., Inganäs, O. & Manca, J. V. Relating the open-circuit voltage to interface molecular properties of donor:acceptor bulk heterojunction solar cells. Physical Review B 81, (2010)] For all cavity enhanced SubNc/C₆₀ devices V_{OC} 's extrapolated to 0 K are indeed found to be similar (Supplementary Figure S8)." (Lines 262-266)

1. Vandewal, K., Tvingstedt, K., Gadisa, A., Inganäs, O. & Manca, J. V. Relating the open-circuit voltage to interface molecular properties of donor:acceptor bulk heterojunction solar cells. *Physical Review B* 81, (2010).

Minor point:

Under Experimental methods, the authors write that the electroluminescence measurements were carried out at injection currents equivalent to the short-circuit currents of the solar cells. I guess they refer to "1 sun"? Please add that information.

Answer to Reviewer #2:

Thank you for pointing this out. The text has been updated including the "1 sun" information.

Best regards

Uli Würfel

Reviewer #3 (Remarks to the Author):

In their manuscript, Nikolis et al. investigate organic solar cells in the strong light-matter interaction regime. They observe polariton branches that give rise to a steepened absorption edge while the charge-transfer state remains unaffected, leading to an effective reduction of energy loss. As the authors discuss candidly, using resonant structures lowers the power conversion efficiency but nevertheless might be interesting for homo-tandem / multi-junction cells. This work is a nice example that even without ultra-strong light-matter coupling the energy levels can be modified in a useful way, and it provides a novel angle for organic photovoltaics.

The paper is very well written, and the arguments and conclusions are sound and convincing. It includes a comprehensive set of data that clearly shows the effects of the strong coupling regime, with all dependencies that are of interest. I have only some minor comments that should be addressed before publication:

1) The present device area is only a few mm^2 . Can the concept be realistically expanded to larger devices, given that already small changes in thickness or homogeneity can lead to a pronounced shift of the resonance wavelength of the cavity?

Answer to Reviewer #3:

When considering large area devices, we can take into account the processing methods adopted by companies working on OLEDs nowadays. Fabrication of organic electronic devices with vacuum processing has proven to provide the thickness accuracy needed for optimized interference effects, yielding maximized outcoupling efficiencies in the case of OLEDs. Given the fact that at this moment, the concept does not yet result in efficiency increases for single junction cells, we decided to further abstain from commenting on this

issue. This issue will however indeed be of importance once this concept is used to improve light harvesting and efficiency in, for example tandem cells.

2) Can the authors explain the additional modulation as a function of angle in Fig. S4 top right?

Answer to Reviewer #3:

We thank the Reviewer for noticing this. After repeating the angle-resolved EQE measurements the observed modulation disappeared, implying that it was actually a measurement and interpolation artefact. This was caused by the single measurement point at 10 degrees having a lower intensity leading the interpolation script that we used to produce a seeming fluctuation in signal.

We have updated Figure S4 in SI.

3) The real Stokes shift can be approximately obtained from the peak centers of absorption and PL peaks only for thin layers without significant reabsorption. Is this condition met for the data in Fig. S1? The Stokes shift should also be given in meV (not only nm).

Reply to Reviewer #3:

Taking care to avoid significant reabsorption, we had measured the absorbance and PL spectra shown in Figure S1 using ultrathin layers of 12 nm and 15 nm, for SubNc and Cl₆-

PhOSubPc respectively. However, the thicknesses were not mentioned in the caption of Figure S1 in the first version of the Supplementary Information (SI). In the updated version of the SI the caption of Figure S1 reads as follows:

“Figure S1. Normalized thin film absorbance and photoluminescence spectra for a) chloroboron subnaphthalocyanine (SubNc) film of 12 nm, and b) hexachloro phenoxy subphthalocyanine (Cl₆-PhOSubPc) film of 15 nm. Inset pictures show the molecular structure of each organic absorber. The peak wavelengths for absorbance and PL are also indicated, and their difference yields a Stokes shift of 16 nm (40 meV) for SubNc, and 25 nm (86 meV) for Cl₆-PhOSubPc.”

Following the suggestion that the Stokes shift should be given also in meV, the caption of Figure S1 was also updated in this regard. Moreover, in the main text:

“SubNc (donor, E_{opt} = 1.73 eV) and Cl₆-PhOSubPc (acceptor, E_{opt} = 2.08 eV) are ideal candidates to demonstrate SC in OSCs, since they both exhibit strong absorption at photon energies close to their E_{opt}, and Stokes shifts of only 16 nm (40 meV) and 25 nm (86 meV), respectively (Supplementary Figure S1).”

Finally, the Figure 2 of the main text was updated including the Stokes shift values in parentheses, while in the caption of the Figure 2 is now written:

“...c) Comparison between EQE and EL peaks of the LP peak for various d. The coloured numbers (in nm and meV) denote the Stokes shift in each case. The minimal Stokes shifts confirm that the investigated devices operate in the strong coupling regime.” (Line 180)

4) Line 340: Should be 0.52 eV instead of 0.52 meV.

Reply to Reviewer #3:

Thank you. The text has been corrected accordingly.

5) Line 392: The bottom Ag(25) layer is missing in the description of the SC layer stack.

Reply to Reviewer #3:

Thank you. The text has been corrected accordingly.

REVIEWERS' COMMENTS:

Reviewer #1 (Remarks to the Author):

My comments were fully addressed by the authors. I believe that the current version of the manuscript is suitable for publication.

Reviewer #2 (Remarks to the Author):

The authors have addressed all points raised by the reviewers in a thorough manner. I do not have any further critical comment and suggest acceptance of this manuscript for publication.

Best regards,
Uli Würfel

Reviewer #3 (Remarks to the Author):

The authors addressed the referees' points satisfactorily in their rebuttal letter and further improved the manuscript and supporting information, making it ready for publication in Nature Communications. Although the outcome is that the impact of strong-coupling on the performance of current single-junction cells seems to be overall detrimental, this aspect needed such high-quality, diligent study and still might find its application in future OPV tandem or multi-junction devices.